# Harnessing the Power of Radiotherapy for Lung Cancer: A Narrative Review of the Evolving Role of Magnetic Resonance Imaging Guidance

**DOI:** 10.3390/cancers16152710

**Published:** 2024-07-30

**Authors:** Sarah Hsin Cheng, Shao-Yun Lee, Hsin-Hua Lee

**Affiliations:** 1Department of Clinical Education and Training, Kaohsiung Medical University Hospital, Kaohsiung Medical University, Kaohsiung 807, Taiwan; 1120246@ms.kmuh.org.tw; 2Department of Medical Education, Taichung Veterans General Hospital, Taichung 407, Taiwan; inty281@vghtc.gov.tw; 3Department of Radiation Oncology, Kaohsiung Medical University Hospital, Kaohsiung Medical University, Kaohsiung 807, Taiwan; 4Ph.D. Program in Environmental and Occupational Medicine, Kaohsiung Medical University and National Health Research Institutes, Kaohsiung 807, Taiwan; 5Department of Radiation Oncology, Faculty of Medicine, School of Medicine, College of Medicine, Kaohsiung Medical University, Kaohsiung 807, Taiwan; 6Center for Cancer Research, Kaohsiung Medical University, Kaohsiung 807, Taiwan

**Keywords:** lung cancer, MR-Linac, magnetic resonance imaging (MRI), adaptive radiotherapy, imaging-guided radiotherapy (IGRT), stereotactic body radiotherapy (SBRT), artificial intelligence (AI), stereotactic ablative body radiotherapy (SABR)

## Abstract

**Simple Summary:**

MR-Linac is a novel magnetic resonance imaging (MRI)-guided radiotherapy (IGRT) that combines MRI with a linear accelerator (Linac). Although radiation therapy (RT) for lung cancer has traditionally been managed with a computed tomography (CT)-based workflow, an MR-Linac-based workflow would be able to address the many limitations of current practice. This narrative review summarizes the latest developments in MR-Linac lung cancer treatment, as well as its boundaries. Future research directions are also highlighted.

**Abstract:**

Compared with computed tomography (CT), magnetic resonance imaging (MRI) traditionally plays a very limited role in lung cancer management, although there is plenty of room for improvement in the current CT-based workflow, for example, in structures such as the brachial plexus and chest wall invasion, which are difficult to visualize with CT alone. Furthermore, in the treatment of high-risk tumors such as ultracentral lung cancer, treatment-associated toxicity currently still outweighs its benefits. The advent of MR-Linac, an MRI-guided radiotherapy (RT) that combines MRI with a linear accelerator, could potentially address these limitations. Compared with CT-based technologies, MR-Linac could offer superior soft tissue visualization, daily adaptive capability, real-time target tracking, and an early assessment of treatment response. Clinically, it could be especially advantageous in the treatment of central/ultracentral lung cancer, early-stage lung cancer, and locally advanced lung cancer. Increasing demands for stereotactic body radiotherapy (SBRT) for lung cancer have led to MR-Linac adoption in some cancer centers. In this review, a broad overview of the latest research on imaging-guided radiotherapy (IGRT) with MR-Linac for lung cancer management is provided, and development pertaining to artificial intelligence is also highlighted. New avenues of research are also discussed.

## 1. Introduction

Lung cancer is one of the most common cancers in the world and a leading cause of cancer death for both sexes [1]. Lung cancer has two main subtypes: small-cell lung cancer (SCLC) and non-small-cell lung cancer (NSCLC). Despite great strides having been made in lung cancer treatment over the past few decades, outcomes still appear dismal. One major treatment modality is radiotherapy (RT), which is used with curative intent for early-stage NSCLC, in combination treatment with systemic therapy during advanced stages, and in palliative treatment in the terminal stage. As imaging technology advances and demands for more precise RT increase, magnetic resonance imaging (MRI)-guided radiation therapy (MRgRT) has been introduced.

The most prominent example of MRgRT is MR-Linac, which combines an MRI scanner with a linear accelerator (Linac). There are currently two clinically operable MR-Linac systems that are commonly used [2]: the 0.35 tesla (T) ViewRay MRIdian and the 1.5T Elekta Unity MR-Linac [3]. Although MR-Linac’s high capital investment still limits its accessibility, its utilization is rapidly expanding around the globe [4,5]. From 2014 to 2020, 0.35T MR-Linac usage in the United States steadily increased, with an annual growth rate of 59.6%, and a large proportion of that growth was seen in on-table adaptive planning and ultrahypofractionation dose scheduling (UHfx), which includes stereotactic body radiotherapy (SBRT) or the so-called stereotactic ablative body radiotherapy (SABR) [4]. A comparable pattern of usage was also reported in an analysis of both European and Asian usage of the 0.35T MR-Linac from 2015 to 2020 [5]. In both analyses, lung cancer was among the most treated tumor types, along with abdominal and pelvic tumors [4,5]. 

Our institutional experience also reflects a similar pattern of usage. We are a medical center located in a metropolis in southwestern Taiwan. A 0.35 T MR-Linac has been commissioned since 2020, and from June 2020 to December 2024, 594 treatment courses were completed, over half of which were SBRT. Lung cancer was among the common malignancies to be treated, and as demand for lung cancer SBRT continues to grow [6], so will MR-Linac use in lung cancer. This paper provides a broad overview of the latest in MR-Linac lung cancer treatment while also highlighting new avenues of research in the area.

## 2. A Brief Overview of the Linear Accelerator’s Development

Figure 1 presents an illustrated timeline of major milestones in the development of Linacs. In 1928, Wideröe proposed the concept of a Linac in a paper titled “On a New Principle for the Production of Higher Voltages” [7]. Although the proposal turned out to be impractical for clinical application, it remained a forerunner of modern Linacs. It was not until 1953 that the first clinical Linac was installed and patients began being treated at Hammersmith Hospital in London [7]. In the subsequent years, various modifications were made to clinical Linacs. For example, the concept of conformational RT and the multileaf collimator (MLC) was first developed by Takahashi in Japan in the 1960s [7]; nevertheless, owing to difficulties in operating the control system, the MLC’s adoption elsewhere would have to wait until later years. By the 1970s, the Linac had established itself, with both the low-energy and multimodality systems being widely available from a number of manufacturers [7]. In the 1990s, David Jaffray developed cone-beam CT (CBCT), which push-started the concept of imaging-guided radiotherapy (IGRT) [8], and around the same time, adaptive RT was first proposed by Yen et al., which we discuss in detail in later sections [3,7]. Although the development of these techniques has mitigated side effects to some extent, there are inadequacies in managing dose distribution [9]. Finally, in 2004 and 2005, Raaymakers et al. [10] and Raaijmakers et al. [11] published the first papers on the feasibility of combining MRI with a Linac, initiating the development of MR-Linac [3]. As mentioned earlier, ViewRay MRIdian and Elekta Unity are two widely adopted systems, and the third system, Aurora RT, which utilizes a 0.5T magnet and 6MV Linac system, recently received U.S. Food and Drug Administration (FDA) approval in 2022 [3]. It boasts a large bore space, thereby allowing for the iso-centric positioning of large tumors [12]. In addition, a prototype is being developed in Australia, with a machine designed to combine a 1.0 T magnet with open-bore MRI and a 6MV Linac [13], and due to its open-bore design, the Australian prototype is versatile and can direct beams at different directions. This version is set to undergo a feasibility and safety study (ACTRN12621000418875).

## 3. Advantages of MRgRT in Lung Cancer Treatment

Figure 2 provides a brief overview of the current RT workflow for lung cancer and its various limitations. Thoracic MRI traditionally plays a limited role in lung cancer treatment. This is because, compared with CT, thoracic MRI tends to contain a poor signal-to-noise ratio secondary to intrafractional respiratory and cardiac motions, as well as low tissue density of the lung parenchyma [14,15]. Various guidelines, including those of the National Institute of Clinical Excellence (NICE), the National Comprehensive Cancer Network (NCCN), and the American College of Chest Physicians (ACCP) therefore recommend MRI use only during the evaluation of suspected chest wall invasions or superior sulcus tumors [16,17,18], with the only other major MRI application in lung cancer management being for the detection of brain metastasis [14]. However, MRgRT has much to offer and could potentially revolutionize lung cancer therapy. In the following section, we discuss the advantages of MRgRT over the current CT-based workflow. A summary of MRgRT advantages can also be found in Figure 2.

### 3.1. Superior Soft Tissue Visualization

Compared with CT imaging, an MR scanner has an inherently superior soft tissue visualization capability [14], which has the benefit of increasing delineation precision around targets and organs at risk (OARs) that have traditionally been challenging to visualize with CT alone, even with the use of a CT OAR atlas. Examples of such structures include the brachial plexus, esophagus, heart, and spinal cord. For these structures, an approach combining CT and MRI is already being recommended [19]. In particular, MRI could be useful in reducing cardiac exposure to irradiation.

The latter has become a topic of growing interest in lung RT ever since the landmark Radiation Therapy Oncology Group (RTOG) 0617 trial demonstrated that radiation exposure to the heart correlates with poor post-RT survival [20], with subsequent analyses finding differences in radiosensitivity among different cardiac substructures. The idea of cardiac substructure constraint was thus proposed; however, clinically, the mean heart dose is still the most commonly considered factor in lung SBRT planning. MRgRT could help cardiac substructure sparing become feasible [21]. Van der Pol et al. conducted a small study of 34 patients with stage II to IV lung cancer treated with SBRT [21]. For each subject, both cardiac-sparing and non-cardiac-sparing MR-Linac treatment plans were created and compared. It was found that 47% of the patients needed cardiac sparing and that a successful cardiac-sparing MR-Linac treatment plan could be created for 62.5% of these patients without compromising doses to the tumor or other OARs.

Similarly, in terms of target delineation, centrally located primary lung tumors and metastatic lymph nodes are also challenging for CT to visualize alone, as we discuss in detail in later paragraphs. To date, studies in pelvic, central nervous system, and head and neck cancers have demonstrated smaller and more accurate target contouring with the addition of MRI than with CT-based delineations alone [22,23,24], while similar studies are sparse in lung cancers due to the absence of suitable thoracic MRI sequences [14,25]. Despite this, a clinical need for a more precise delineation still exists in lung RT, particularly in cases with tumors invading the mediastinum, abutting parenchymal lung changes, or invading major arteries [14]; to date, an accurate assessment of disease extent in such situations remains challenging. Although the addition of an ^18^F-labeled fluoro-2-deoxyglucose positron emission tomography (F-18-FDG PET) scan to CT has been found to reduce inter-observer variations in challenging cases, the spatial resolution of a PET scan, along with the healthy and cancerous tissue contrast, is still inferior to that of MRI [14,26]. Therefore, the International MR-Linac Consortium, which is a collaboration between researchers and commercial partners to introduce MR-Linac, is working on refining MRI sequencing for IGRT [27]. Additionally, two recent small studies both demonstrated T1-weighted DIXON sequences to be the preferred MRI sequence for thorax imaging and contouring in healthy non-patient volunteers due to its image contrast resembling that of planning CT, which makes image interpretation and subsequent registration more straightforward [25,28]. Naturally, larger studies with patient volunteers are needed to validate these results, and the optimization of MRI sequences for lung IGRT is a prerequisite to carrying out any subsequent high-quality studies regarding MRgRT in lung cancer management.

Thus far, studies in MR-based planning have shown that diffusion-weighted MRI (DW-MRI)-based gross tumor volume (GTV) correlates with CT-based GTV plans in lung RT [29]; however, there is a paucity of clinician experience in target delineation with lung MRI, as reflected by studies on inter-observer delineation variability described below. Karki et al. compared inter-observer variations between MRI and PET-CT-based GTV, with MRI appearing to have an acceptable yet greater inter-observer variability than PET-CT [26]. In another recent study that evaluated the inter- and intraobserver variability of MR- versus PET-CT-based GTV for lung cancer SBRT, greater inter-observer variability was also found between MR-based GTV plans, despite significant intraobserver agreement [30]. Both teams of investigators attributed the inter-observer variation to a lack of observer experience with MR-based lung GTV delineation. As a growing number of MR-Linac systems become available globally, this variability might reduce as more clinicians gain experience with the MR-based delineation of lung tumors.

As for position verification, the current clinical standard is CBCT imaging with online target matching to decrease positional error and ensure consistency in the dose delivered [14]. Compared with traditional fan-beam CT, CBCT induces significantly less radiation exposure, especially since the development of the low-dose CBCT protocol [31]. Nevertheless, a CBCT image is also subject to higher degrees of scatter and artifact, rendering it inadmissible for subsequent adaptive planning; thus, the generation of synthetic CT (sCT) using an artificial intelligence (AI) model was developed to address this issue [31]. Unfortunately, CBCT-generated sCT has the same limitation as planning CT, i.e., poor soft tissue visualizations, and MRgRT could be advantageous in this regard. Current AI technologies are already able to generate sCT from MR images of the thorax with a good dose accuracy and image quality, without inter-scanner variability or radiation exposure [32]; additionally, MR-derived sCT has demonstrated good overall agreement with conventional CT in lung cases [33].

### 3.2. Daily Adaptive Capability

MR-Linac is a major advance in IGRT because of its adaptive capabilities [34]. Thus, any discussion on MR-Linac requires a brief introduction to adaptive radiotherapy (ART). ART was first proposed by Yan et al. at the turn of the century [35] as a means of simply refining the planning target volume (PTV) to account for uncertainties during delivery [34]. Over the years, ART has evolved to use advanced imaging systems to visualize the anatomical and functional changes in patients as they progress through their treatment, and modifications are made accordingly. The ART concept believes that a baseline simulation is just a snapshot of our body at the time of the simulation, and regardless of how precise it is, throughout the treatment course, changes in the patient’s body inevitably occur. These include tumor shrinkage, as well as normal day-to-day changes in body anatomy, such as bowel peristalsis and bladder filling. In short, the treatment plan devised on the baseline simulation will likely not fit the patient on the day of treatment, and continuous usage of the outdated simulation plan will cause subsequent problems, such as suboptimal dosing and radiation toxicity to healthy tissues. In a recent phase II clinical trial, the LARTIA study, the benefit of ART was demonstrated [36]. LARTIA showed that ART in the form of weekly replanning CT scans led not only to a low rate of marginal failure but also to a low rate of toxicity at a median follow-up of 20.5 months. Although a definitive conclusion will have to wait until larger phase III studies, this result provides impetus for further ART research in lung cancer.

In general, the scientific community supports the concept of ART, and over the years, ART has become more sophisticated, with MR-Linac being its latest breakthrough. Lamb et al. reported that there were nine centers equipped with onboard MRgRT in 2017 [37] and published a technical report on the process of ART. Currently, there are two commonly seen classes of ART technologies [38]: (1) offline ART, where verification and adaptive planning are scheduled between fractions, which means that the adaptive process can take days, and (2) online ART, where verification and adaptive planning are carried out immediately before treatment delivery, with the time between the adaptive process only taking minutes (Figure 2). The Adaptive Radiation Therapy Physician Guidelines were published in 2022 with recommendations from experts from the USA, UK, and Korea [39]. The third type of ART, real-time ART, can reduce the adaptive process to seconds, although this is rare in clinical practice and falls outside the scope of our discussion.

Prior to the advent of MR-based Linac systems, only offline ART was available clinically; thus, MR-Linac was revolutionary in RT because it enables online ART. Compared with the LATRIA study that utilized weekly adaptive CT scans, MR-Linac has the capability to offer rapid and automated online plan adaptation prior to treatment delivery each day. With MR-Linac’s daily adaptive capability, there is a possibility to realize the full potential of adaptive planning while minimizing concomitant radiation exposure [14]. Indeed, a recent study by Kang et al. supported this view by demonstrating that lung SBRT with MR-Linac versus a CT-guided modality could decrease the radiation field without lowering the local control rate and survival rates at 1 year [40].

### 3.3. Real-Time Target Tracking

Prior to MR-Linac, the only option for real-time tracking was the application of internal fiducial markers [14], which required the fiducial marker to be positioned at or near its target. A kilovoltage imaging system was then used to image the marker. In patients with locally advanced diseases and requiring multiple fiducial markers, or in patients with protracted treatment courses, the practice could cause significant radiation exposure [14]. However, MR-Linac systems could also offer intrafractional motion detection, yet without additional ionizing radiation.

For thoracic imaging and RT, a major source of intrafractional motion is respiratory motion. Currently, respiratory motions are accommodated through the expansion of the clinical target volume (CTV)–planning target volume (PTV) margin [41]. It is important to note that, although 4D-CT could account for structural changes with time, it does not account for real-time daily motion management. This is because 4D-CT is conducted at simulation, where the machine studies tumor motion through a patient’s respiratory cycle and calculates an internal target volume (ITV) based on the union of all tumor positions during the imaging process; however, 4D-CT does not account for the changes that occur during RT delivery. In fact, a study using real-time MRI found that ITV size often varies more than previously considered, suggesting significantly higher inter- and intrafractional variations in breathing amplitude than accounted for during the current ITV-based planning process [42]. The real-time anatomical and dosimetric information can enable intrafractional plan adaptation, ensuring the intended daily dose is delivered. This approach may allow for smaller PTV margins and further reduce the dose to healthy tissues [43].

While the current literature shows that ITV-based SBRT is adequate with a tolerable toxicity profile for most patients, there is still much room for improvement, especially in cases where minimizing the treatment margin is crucial [42]. This includes patients with central lung tumor(s) and patients with erratic breathing patterns; for these populations, a more precise treatment option with real-time target tracking that minimizes intrafractional variation is necessary.

Currently, the 1.5T MR-Linac utilizes MLC tracking, for which studies have demonstrated technical feasibility and improved dose accuracy [44,45]. However, the 0.35 T system uses automatically gated beam delivery with cine MRI; this is often coupled with patient self-sustained breath-holds (BHs) [46], which can be monitored via real-time imaging, though this is still subject to patient compliance and individual respiratory function. To address this issue, the Active Breathing Coordinator (ABC) was invented to improve lung volume reproducibility, thereby lowering intrafractional variation [47]. In radical NSCLC RT specifically, the ABC has been shown to reduce the treatment margin [48]. The ABC has been used in SBRT with conventional Linacs for many years. Recently, the first MR-ABC protocol was established by Kaza et al., and their results showed that the ABC, in lung cancer MRI, could lead to good lung volume reproducibility [49]. This means that the ABC could soon be used for gating purposes on MR-Linac to further reduce uncertainties in BH techniques. Alternatively, an AI algorithm has also been developed that could effectively predict tumor motion in 0.35 T MR-Linac machines, enabling the future implementation of MLC tracking [46].

### 3.4. Early Assessment of Treatment Response

In the current workflow, there is no early assessment of treatment response (Figure 2). However, in an MR-based workflow, in addition to the conventional qualitative MRI, clinicians will be able to use quantitative MRI (qMRI) to quantitatively map out a tumor’s biological information, thereby evaluating changes early in the treatment in a more timely manner to predict responses. Various qMR techniques and associated non-invasive imaging biomarkers have been identified, with two of the most investigated MR techniques being diffusion-weighted imagining (DWI) and dynamic contrast-enhanced (DCE) MRI [14,50]. Table 1 contains a list of the qMR techniques that have been investigated in the context of lung cancer.

DWI MRI calculates the apparent diffusion coefficient (ADC), which is associated with the magnitude of diffusion within a tissue [50]. When cells die and the cell density reduces, diffusion increases, and an increase in the ADC is observed. It has been reported that the ADC value increases during chemoradiation in NSCLC [53]. However, compared with cancer of other organs, where a greater increase in the ADC value might predict responders [50], lung cancer studies are scant due to the traditionally limited role of thoracic MRI, and of those few studies available, the results are mixed [50,54]. There is even a small study that suggests that the ADC value increase at 1 month is associated with local failure in NSCLC [55]. However, importantly, researchers have shown that the ADC biomarker is able to predict responses before morphological changes can be seen [54]. This has important implications for regimen design, as patients may be able to switch or terminate anticancer drugs based on early biological marker changes, thereby avoiding unnecessary systemic toxicity or delayed time in receiving more effective treatment options [54]. In fact, ADC data have been found to improve the performance of the NSCLC survival prediction AI model [63], further substantiating the importance of the ADC as a biomarker for treatment response.

In contrast, DCE-MRI measures tissue perfusion and capillary permeability [50]; thus, within the tumor microenvironment, where an abundance of abnormal microvasculature can be found, there tends to be an enhancement in DCE-MRI. Although the kinetic parameters of capillary permeability have been linked to tumor response to RT in some cancers [64,65,66], the data for lung cancer remain scarce, and of the limited studies, DCE-MRI seems promising for the prediction of treatment response following concurrent chemoradiation (CCRT) in NSCLC [58] and stereotactic radiosurgery (SRS) for metastatic brain tumors from primary lung cancer [59].

Further investigation is still needed. A major downside to the DCE MRI technique is the need for contrast agents, which come with their own risk, and as a result, options for non-contrast-enhanced MRI methods are being explored [67]. Oxygen-enhanced MRI is also being explored for the detection of regional hypoxia in NSCLC [62], as this might predict tumor regions refractory to RT, and additional treatment strategies might therefore be employed. As various qMRI techniques are being explored, it is also important to validate the techniques against one another. For example, it has been found that, although both intravoxel incoherent motion (IVIM) and DCE-MRI could be used to measure metastatic brain tumor blood volume post-SRS, their measurements do not correlate. In a comparative study by Kapadia et al., it was shown that while DCE-MRI showed a reduction in tumor blood volume, IVIM showed the opposite, or an increase in blood volume [68]. Therefore, until qMRI techniques have been well validated, the interpretation of single qMRI techniques should be performed with caution, with multiple parameters being considered when necessary. In short, numerous qMRI techniques have potential as biomarkers for early treatment response, although more data on lung cancer are still required.

### 3.5. Combining Biological Targeting with Conventional ART

As discussed, the images produced by the above qMRI techniques are called biological images. They reveal various biological data, ranging from metabolic to functional data that reflect a tumor’s biological heterogeneity. These data can then be used to establish a dose distribution, which was validated to be different from that of conventional anatomical imaging [69,70]. These biological image-based dose distributions then allow clinicians to visualize tumor radiosensitivity [71], which is invaluable for treatment response prediction, as previously discussed, while also helping to reduce toxicity and preserve function. An example is the use of functional information for normal lung avoidance. Quantitative gas MR has been developed, and it could allow us to directly visualize pulmonary ventilation and perfusion [72]. Areas of pre-existing pulmonary dysfunction could be identified, and by incorporating functional information into RT planning, we could perform accurate dose escalation to areas of low pulmonary function while decreasing radiation exposure to areas of highly functioning normal lung tissue [73]. Although single-photon emission computed tomography (SPECT) and PET-CT could also visualize lung function, gas MR has the added advantage of a superior spatial and temporal resolution without ionizing radiation exposure [72]. Unfortunately, the only gas MR-based functional lung avoidance RT clinical trial in recent years was terminated due to insufficient accrual (NCT02002052).

Just as conventional ART emphasizes morphological changes during RT, the biological heterogeneity of a tumor is also dynamic, making it vital that biological images also undergo adaptive planning, which has become known as biological image-guided adaptive RT (BIGART) [50]. Before MRgRT, CT-based BIGART had already demonstrated feasibility and tolerability in clinical trials [74,75]. Unfortunately, it is still difficult to realize in practice due to logistical challenges [50]. For example, because planning CT and functional images are not acquired at the same time, it is difficult to compare the images; thus, the validity of image comparison and the accuracy of image registration are called into question [73]. In contrast, MR-Linac has the potential to perform daily biological imaging and adaptive planning concurrently, which fully realizes the potential of BIGART [50]. However, before BIGART can become a clinical routine, many requirements still need to be fulfilled. For example, Van Houdt et al. believe that issues such as investigating daily changes in qMRI values, choosing the most suitable qMRI techniques as biomarkers, and standardizing qMRI protocols all need to be dealt with first [50], and for each of the above steps, large, multicenter clinical trials are needed for validation.

### 3.6. AI and Machine Learning

AI, especially its subset, machine learning (ML), is revolutionizing radiology by enhancing image analysis and reducing diagnostic errors [76]. AI is transforming radiology by optimizing workflows and improving non-interpretative tasks [77]. Integrated with Natural Language Processing (NLP), AI automates the triage of imaging studies, prioritizing urgent cases through electronic health records and expediting patient triage, reporting, and follow-up management [78]. Deep learning (DL), an ML subset, improves report consistency and clarity, enhancing radiology service quality. DL accelerates MRI scanning, harmonizing efficiency and quality, with commensurate progress witnessed in CT and PET image reconstruction.

Understanding the interrelationships between AI, ML, and DL helps to conceptualize each subfield’s contribution and progression within the broader AI narrative [76]. AI provides the foundation, ML enhances AI’s potential by enabling machines to learn from data, and DL further extends these capabilities with neural networks that decipher complex data patterns [76]. Despite AI’s potential, only 30% of radiologists used it clinically by 2021, with many skeptical [79]. Effective AI integration requires supportive infrastructure and workflow redefinition. It may help reduce errors and radiologist burnout [80].

The integration of AI, such as automation into the RT planning pathway, will allow ART to become a reality in clinical practice [81]. In one study, AI-generated data were favored over conventional MRI planning data by the radiation oncologist in the study [82]. Accordingly, we believe that the development of AI-driven ART could speed up the MRgRT workflow and overcome a major barrier to MRgRT implementation.

## 4. MRgRT Clinical Application

Since most MR-Linac systems for lung cancers are used in SBRT treatment [4,5], the following discussion focuses on SBRT. Table 2 contains all ongoing clinical trials involving MR-Linac at the time of the writing of this manuscript.

### 4.1. Central and Ultracentral Lung Tumor

Despite advances in lung cancer treatment with SBRT, treatment for central (CLT) and ultracentral (ULT) lung tumors remains a challenge due to significant toxicity. Here, we adopted the Nordic-HILUS trial’s definition, where a central tumor is defined as a tumor within 2 cm of the proximal bronchial trees, while ULT is defined as touching or within 1 cm of the mediastinum, esophagus, and proximal bronchial tree [83]. Such proximity to vulnerable mediastinal structures, coupled with respiratory and cardiac motions, makes the grade 5 toxicity risk with SBRT as high as 15.4% [83]; consequently, the American Society of Radiation Oncology (ASTRO) deems the toxicity risk to negate any treatment benefit and therefore discourages SBRT to ULTs [84]. Here, MRgRT’s strength in gated dose delivery and daily plan adaptation may be useful in mitigating the toxicities. In fact, Ligtenberg et al. demonstrated that clinicians could reduce CLT treatment toxicities on MR-Linac by lowering the OAR doses of ITV-based plans with mid-position-based planning, a method based on the time-weighted mean position of the tumor [85]. Furthermore, CLT dose accumulation studies have shown that, at 1 cm of the PTV, the OAR dose of online adaptive MRgRT is not significantly different from that of non-adaptive gated proton therapy or even online adaptive proton therapy [86]. Given their similar toxicity profiles, MRgRT presents as a more attractive treatment option than proton therapy due to its lower structural investment.

Current clinical trial results support the use of stereotactic MR-guided adaptive radiation therapy (SMART) in treating CLTs/ULTs. Regnery et al. performed a small prospective cohort study that showed that SMART in ULT treatment was able to protect the OARs while maintaining adequate PTV coverage, and in the subsequent long-term follow-up, they found no statistically significant difference in 2-year OS, PFS, or local progression between the ULT and non-ULT groups [87]. However, significantly more toxicities of ≥ grade 2 occurred in the ULT group, two of which were grade 3 esophagitis and bronchial bleeding, but most patients experienced mild toxicities only. The results showed SMART to be a potentially safe and effective treatment for ULT, although ULT remains a high-risk target area [87].

In contrast, Sandoval et al. also treated ULT and CLT with SMART [88]. At 1 year, OS was 82%, the local control rate was 87%, and PFS was 54%, with most patients experiencing grade 1 and 2 toxicities, while only two patients experienced grade 3 toxicities. Finazzi et al. also reported a promising 1-year local control rate (95.6%) and low high-grade toxicity risks (30% for ≥ grade 2 and 8% for ≥ grade 3 toxicities) in their trial of high-risk lung tumor patients treated with SMART [89], where the inclusion criteria were not limited to patients with central tumors but also encompassed patients with previous thoracic RT histories and interstitial lung diseases. Despite optimistic results, data from larger clinical trials are still needed before definitive conclusions can be drawn. Currently, a prospective phase I dose escalation trial, MAGELLAN, is underway [90]. It will enroll primary and secondary ULT patients with tumor size ≤ 5 cm to be treated with a 0.35 T MR-Linac. Dose escalation will begin at 10 × 5.5 Gy and reach a maximum of 10 × 6.5 Gy under the guidance of a time-to-event continual reassessment method. MAGELLAN will establish the safe dose for UCT treatment with SMART, which will guide further research in the area.

### 4.2. Early-Stage Lung Cancer

In RT, the current standard treatment for early-stage inoperable NSCLC is SBRT, with two methods of SBRT delivery available: over multiple fractions or a single fraction. Owing to concerns about missing the tumor target and toxicity to OARs when administering treatment in one large single dose, single-fraction delivery is not popular among clinicians [91], and while internal fiducial markers could potentially be employed to increase accuracy, issues with radiation exposure, as discussed earlier, still exist. Nevertheless, there is no clinical evidence showing that single-fraction SBRT is inferior to its multiple-fraction counterpart. In fact, two randomized phase 2 trials showed that there are no short-term survival or toxicity differences between single- and multiple-fraction treatments in early-stage NSCLC [92,93]. At a five-year follow-up, there still appeared to be similar toxicity rates, tumor control rates, and median survival times [93]. However, the quality of life measurement, such as social functioning and dyspnea, appeared to be better in the single-fraction arm treatment in one clinical trial [92]. In addition, single-fraction SBRT might potentially be more immunogenic than multiple-fraction SBRT due to a potentially greater lymphocyte-sparing effect, although further clinical evidence is pending [94].

During the COVID-19 pandemic, there was a renewed interest in single-fraction SBRT due to the need to reduce patient exposure to the hospital environment [94]. At present, however, with MR-Linac’s real-tracking ability, single-fraction SBRT could become a standard, although problems with its implementation, especially its potentially lengthy treatment time, still need to be resolved. Finazzi et al. recently reported their institutional experience with treating lung tumors with single-fraction, gated breath-hold (BH) SBRT on a 0.35T MR-Linac [95]. Of the 10 patients treated, a median in-room procedural time of 120 min and a median treatment time of 39 min were reported, which is significantly longer than the 15 min treatment time required for flattening-filter-free (FFF) volumetric-modulated arc therapy (VMAT), a faster and more conventional method of performing SBRT [96]. The total procedural time included two breath-hold (BH) 3D MR scans, with one at the start of the treatment day, which was immediately followed by plan editing and adaptation; then, the patients underwent a second scan during mid-treatment with optional readaptation and a break if needed. Their final analysis revealed that the resulting BH patterns among the patients were variable and ultimately found to have no influence on GTV coverage with MR-Linac’s real-time tracking. Although the on-table adaptation improved PTV coverage, it had no influence on GTV doses. Finazzi et al.’s result raised concerns over the increased treatment time and labor-intensive treatment procedure [95].

In comparison, Chuong et al. reported another case of peripheral early-stage NSCLC treated with mid-inspiration BH single-fraction SBRT on MR-Linac [97]; however, their reported procedure time was 40 min, while the treatment time was only 17 min. This shorter treatment time was attributed to a few modifications made to the method of Finazzi et al. Firstly, they eliminated on-table adaptation as they deemed it to be clinically unnecessary for peripheral tumors and Finazzi et al. reported it to have unclear clinical benefits. Secondly, they utilized visual biofeedback respiratory gating, which involved a monitor in the treatment room that showed the patient real-time cine-MR imaging with the target position and treatment boundary. By watching the monitor, the patient was able to time their breath-holding according to the target location within the boundary. Visual biofeedback was so effective in achieving the correct BH position that mid-treatment breaks were eliminated from the protocol. The clinical usefulness of visual biofeedback was also recently corroborated by a Korean study [98]. The small study of 15 patients demonstrated that a similarly concepted visual guidance patient-controlled (VG-PC) respiratory-gated MR-IGRT was able to reduce the total treatment time by 37.6% compared with free-breathing. Our institutional protocol includes on-table adaptation with or without breath-holding. Moreover, our data suggest that procedural optimization over time will significantly reduce the MR-Linac total in-room treatment time.

### 4.3. Locally Advanced (LA)/Oligo-Progressive Lung Cancer

The current standard of treatment for LA NSCLC is platinum-based CCRT, followed by consolidation with immunotherapy, although the prognosis remains poor [99]. Despite major advances in lung cancer treatment, such as immunotherapy, relatively few clinical studies have been conducted in LA NSCLC patient groups compared with metastatic lung cancer patient groups. For example, in the study of the immune checkpoint inhibitor (ICI) and SBRT synergy, no clinical trials specifically for LA NSCLC were completed, and only one small phase II study is currently underway (NCT03589547); consequently, effective treatment for this patient group is urgently required. When MRgRT was first introduced, most studies focused on treating central or ultracentral lung tumors, as previously discussed, but now there is a push toward using MR-Linac for the definitive treatment of LA lung cancer.

In LA-NSCLC, MR-Linac may be proven advantageous for this population due to its powerful soft tissue imaging capability. As discussed earlier, PET scans and CT are not omnipotent, and in cases where there are local invasions into surrounding soft tissues such as the mediastinum or the chest wall, MRI could offer clearer imaging for delineation than CT or FDG-PET [16,17,18]. Even in nodal staging for LA patients, several published studies, including meta-analyses and systemic reviews, agree that DW-MRI has better specificity than CT or FDG-PET [100,101]. Recently, a study that compared two MRI sequences with CBCT on their ability to identify lymph node invasion in LA-NSCLC patients was published, where both MRI sequences (T2w Turbo Spin-Echo and T1w DIXON water-only images) were found to have significantly better image quality and generated more observer confidence than CBCT, despite there being no significant inter-observer variability among the imaging modalities [102].

The development of RT for lung cancer has always been limited by a balance between effective RT dose escalation and normal lung tissue sparing [15]. Retrospective planning studies have shown that the use of MR-Linac for RT treatment could decrease the PTV margin and increase OAR sparing, thereby achieving isotoxic dose escalation and the widening of the therapeutic index [103]. However, the implementation of definitive CCRT using MRgRT for LA lung cancers is labor-intensive and significantly increases treatment time. A standardized protocol has not yet been set up and validated, and only case reports detailing individual institutional workflows have been published.

Recently, a technical report was published by Eze et al. that details a new clinical pathway for a node-positive LA-NSCLC patient with severely compromised pulmonary function and reserve [104], where a 0.35 T MR-Linac was used to deliver a total dose of 48.9 Gy/16 daily fractions with real-time continuous tracking by a 2D MRI during treatment. The treatment was well tolerated, with no grade 2 or above toxicities, and a 3-month follow-up CT scan showed partial remission per RESIST 1.1 criteria. La Rosa et al. also published a case of a stage IV NSCLC patient with oligo-progression and lymph node invasion who was treated with MRgRT [105]. As a result of online ART, the cumulative dose to OARs was reduced by 11.34% (esophagus), 4.2% (proximal bronchial tree), and 5.62% (trachea). These reports demonstrate the clinical feasibility of MR guidance, but both protocols still await validation in the form of large clinical trials.

The current clinical trial NCT03916419 is a single-institution, single-arm, phase II study with a safety lead-in, with the aim of investigating the safety and efficacy of MRgRT in curative-intent CCRT treatment for inoperable LA-NSCLC patients. Its primary end points are the number of participants with dose-limiting toxicities, and local and regional control rates. The study is set for completion in 2025. Both trials will add valuable information to the integration of MRgRT into the LA-NSCLC curative-intent treatment workflow.

## 5. Challenges for MRgRT in Lung Cancer Treatment

Despite its numerous advantages over current CT-based technologies, many limitations related to MRgRT exist. One such challenge is the potential effect of the Lorentz force. During photon irradiation, electrons deposit energy along their pathway, but when these electrons encounter a magnetic field, their path may be deflected by the Lorentz force, which then alters the dose distribution. This is especially significant in areas of heterogenous tissue density such as skin–air or lung–tissue interface regions [11,106]. The Lorentz force at these sites can cause electrons that have already exited the tissue to return to its surface in a phenomenon called the electron return effect (ERE), and in lung RT, ERE can potentially lead to lung tissue toxicities [11,107]. Fortunately, research so far has not found the dose alteration to be clinically significant [44,107]. Moreover, lung RT usually employs low-field imaging for better imaging quality; thus, the ERE is much lower than in high-field imaging [108].

Clinically, there are several methods for MLC calibration or quality assurance [109]. Some challenges related to the MRgRT workflow include the development of a reliable and consistent quality assurance and quality control procedure that is specific to MRgRT [110]. In addition, there is a relative contraindication to treat patients with metallic implants or cardiac devices. Yang et al. recently reported safely treating four patients with a cardiac pacemaker using a 1.5T MR-Linac [111]; similarly, Keesman et al. also published their MRgRT workflow for three prostate cancer patients with a metallic hip prosthesis [112], although both protocols await confirmation through larger studies. Substantial capital investment and training for team members also impede MR-Linac’s widespread use, and finally, the low throughput and the time-consuming adaptation process are also major drawbacks of this novel technology. Most importantly, for individuals that have experienced episodes of claustrophobia during an MRI scan, MRgRT may not be suitable.

Moreover, significant debate remains regarding the prolonged treatment times. Such limitations can impact patient comfort, compliance, and overall treatment efficiency. In our experience, from 2020 to 2023, for non-single-fraction BH SMART, the median in-room procedural time was 37 min, with a median treatment time of 15 min. As discussed in Section 4.2, visual biofeedback respiratory gating was studied by Chuong et al. [97] and reported with good efficiency, and it was examined in another Korean study [98] in which the total treatment time was reduced by 37.6% compared with free-breathing.

Some researchers have taken things further by shortening the entire workflow to one day. A one-step-shop (OSS) service is a procedure well known in the palliative setting that condenses consultation, simulation, and RT delivery all in one day [113] with the goal of shortening overall treatment time, thus improving patient convenience and ameliorating patient discomfort. This is especially important for countries such as the United States and Canada, where patients often must travel great distances to receive RT; therefore, shortening treatment to one day means that time and costs for traveling and lodging could be significantly reduced, further improving the patient adherence rate.

In the past, Brunenberg et al. demonstrated the feasibility of OSS-palliative RT with an MR-based diagnostic protocol and MR simulation [113], and recently, Palacios et al. applied this procedure to single-fraction SMART for small tumors of early-stage lung cancer [114]. The reported median procedural time was 6.6 h, and with procedural optimization, the treatment could be completed in half a day. Good patient satisfaction was reported, with highly accurate gated treatment delivery, as demonstrated by the above 78–100% GTV coverage and 94.4–100% corresponding PTV values. Taken together, these studies demonstrate that MRgRT is capable of performing single-fraction SBRT with high precision, but plenty of room remains for discussing the exact workflow and implementation.

However, as we previously discussed, research efforts are being made to ameliorate these problems. The PUMA trial (NCT05237453) is a prospective, multicenter phase I trial in Germany that will test the clinical feasibility of the two MR-Linac systems in CCRT treatment for LA-NSCLC patients [115]. It will use conventionally fractionated RT with dose escalation and weekly online adaptation. Their primary end points include the successful completion of online-adapted fractions and on-table time. Additionally, AI development and procedural optimization, as team members become more familiar with the new workflow, can significantly increase throughput over time. For example, the manual delineation of OARs represents one of the most time-consuming processes of MRgRT, and AI technologies that can perform both left and right lung auto-segmentations with high accuracy have already been developed [82].

## 6. Conclusions

MR-Linac is a transformative technology that offers significant advantages over the current CT-based systems in treating lung cancer patients. Its superior soft tissue contrast, daily adaptive capability, and real-time image tracking set it apart. Furthermore, MR-Linac enables the integration of biological and anatomical images into the adaptive planning process, enhancing its clinical utility in treating early-stage lung cancer and locally advanced non-small-cell lung cancer (LA-NSCLC).

Despite its impressive benefits, MR-Linac does have limitations. As the demand for lung cancer SBRT grows, the adoption of MR-Linac in lung cancer treatment will become more prevalent. Clinicians will gain more experience and data, leading to improved outcomes. To ensure that expediency does not compromise treatment quality or patient safety, future research should focus on optimizing the workflow procedure. AI will play a crucial role in this advancement, enhancing the precision and efficiency of lung cancer treatment with MR-Linac.

## Figures and Tables

**Figure 1 cancers-16-02710-f001:**
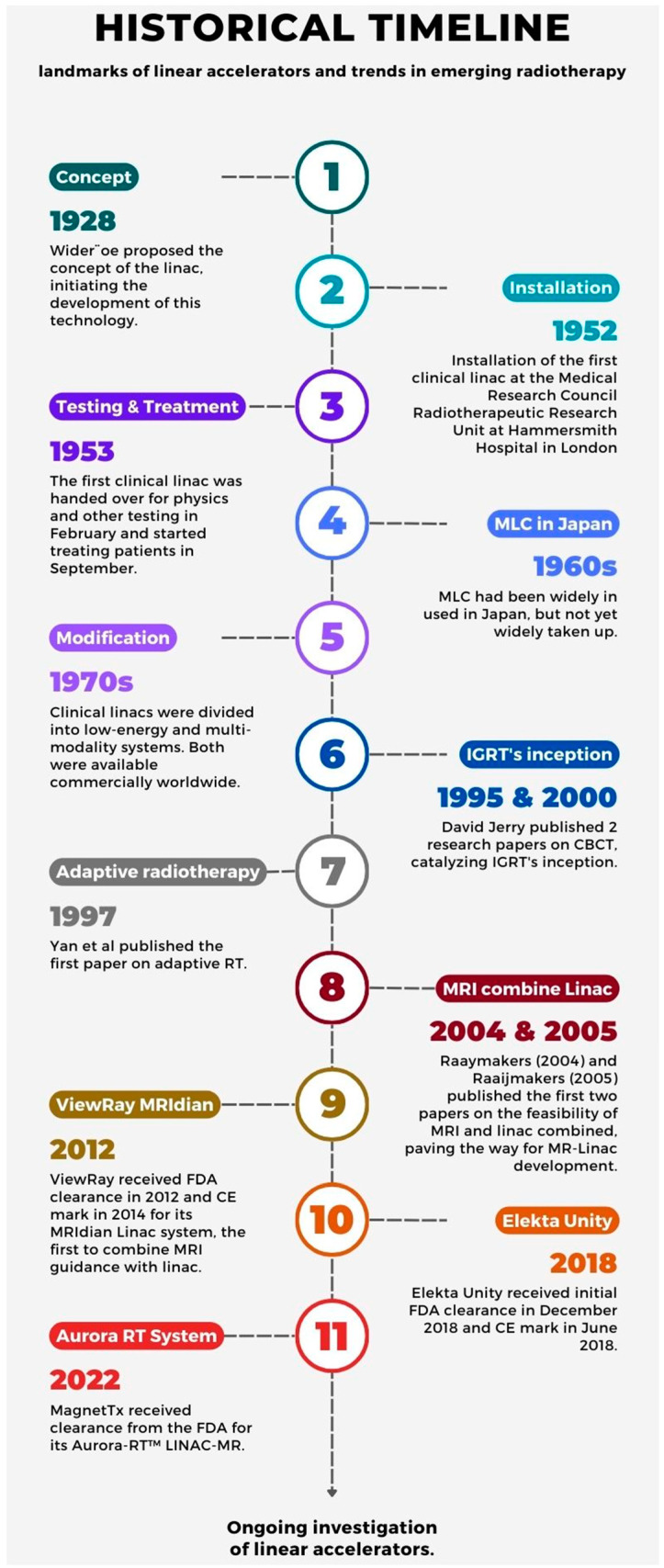
A historical timeline of linear accelerator development and emerging trends in radiation therapy. MLC: multileaf collimators; IGRT: image-guided radiation therapy; CBCT: cone-beam CT; RT: radiotherapy; MRI: magnetic resonance imaging; FDA: Food and Drug Administration.

**Figure 2 cancers-16-02710-f002:**
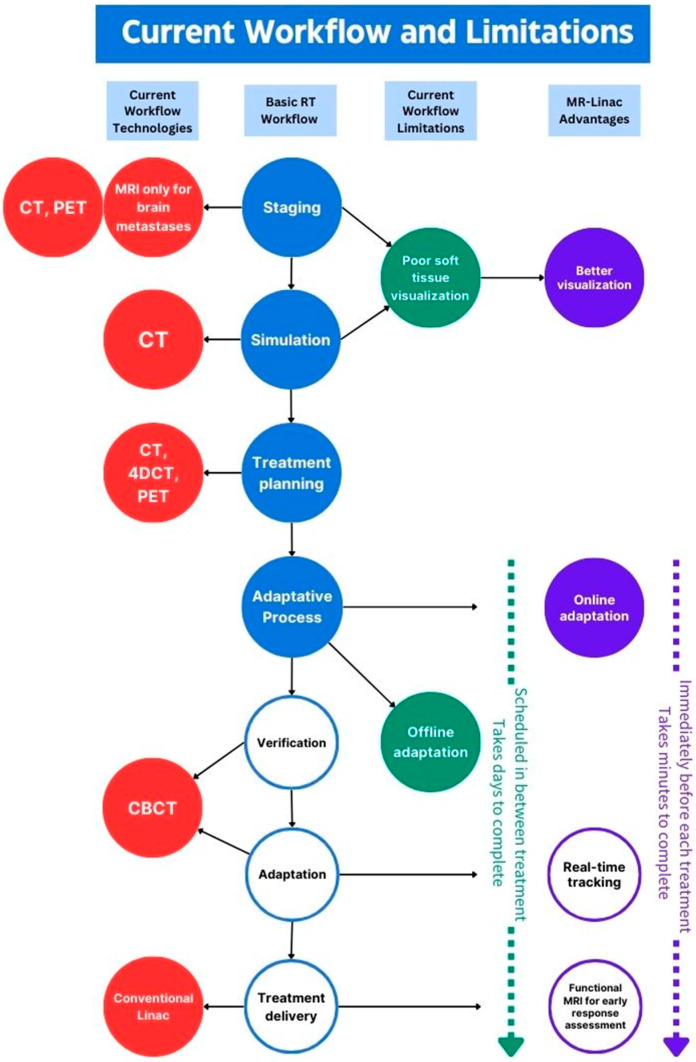
Current radiotherapy workflow, its limitations, and MR-Linac advantages. RT: radiotherapy; MRI: magnetic resonance imaging; CT: computed tomography; PET: positron emission tomography; 4D-CT: 4-dimension computed tomography; CBCT: cone-beam computed tomography.

**Table 1 cancers-16-02710-t001:** Overview of studies on qMRI metrics as predictor of lung cancer treatment response.

Study, Publication Year	MRI Technique	qMRI Metric	Lung Cancer Type	Number of Patients	Follow-up Time Points
Shintani, T., et al., 2017 [51]	DWI	ADC, SUV_max_	NSCLC	14	Pre, 3, 6, 9, 12 months
Chang, Q., et al., 2012 [52]	DWI	ADC	advanced lung carcinoma	14	at regular intervals until the date of death
Weiss, E., et al., 2016 [53]	DWI	ADC	Adenocarcinoma (4) SCC (6)	10	Pre, 3, 6 weeks
Yabuuchi, H., et al., 2011 [54]	DWI	ADC	NSCLC	28	Before and after the first course of chemotherapy
Sampath, S., et al., 2019 [55]	DWI	ADC	NSCLC	13	Pre, 1 month
Sun, Y. S., et al., 2011 [56]	DWI	ADC	Lung cancer	21	Pre, 1, 3, 6 weeks
Seki, S., et al., 2020 [57]	DCE-MRI	Pulmonary arterial perfusion Systemic arterial perfusion Total perfusion	Adenocarcinoma (35) SCC (7) LCLC (1)	43	Pre, every 6 months post-treatment
Tao, X., et al., 2016 [58]	DCE-MRI	BF, BV, MTT, K^trans^, K_ep_, V_e_, V_p_	NSCLC	36	Pre, 1 month
Mehrabian, H., et al., 2017 [59]	DCE-MRI	k_IE_, k_ep_, M_0,I_, M_0,E_, M_0,V_	Primary lung cancer	9	Pre, 1 week, 1 month
Desmond, K. L., et al., 2017 [60]	CEST	APTw, AREX, Lorentzian peak properties	Brain metastases from primary lung cancer	25	Pre, 1 week, 1 month
Gutsche, R., et al., 2022 [61]	Radiomics	Local textural features	Brain metastases from NSCLC	80	Pre, 180 days post-treatment
Salem, A., et al., 2019 [62]	OE-MRI	perfused Oxy-R	NSCLC	23	Pre, post-treatment

Papers were searched on PubMed with search terms “early response” and “radiotherapy” as well as MRI techniques and qMRI metrics mentioned in the table above. Only studies in humans and published in English were included. Reference list of the included papers was checked to identify other relevant papers. DWI, diffusion weight imaging; ADC, apparent diffusion coefficient; SUV_max_, maximum standardized uptake value; NSCLC, non-small-cell lung cancer; Pre, pretreatment; SCC, squamous cell carcinoma; DCE-MRI, dynamic contrast-enhanced MRI; LCLC, large cell lung carcinoma; BF, Blood Flow; BV, blood volume; MTT, Mean Transit Time; K^trans^, endothelial transfer constant; K_ep_, reflux rate; V_e_, fractional extravascular extracellular space volume; V_p_, fractional plasma volume; k_IE_, extracellular water exchange rate constant; k_ep_, efflux rate constant; M_0,I_, M_0,E_, M_0,V_, water compartment volume fractions; IVIM, intravoxel incoherent motion; *f*, perfusion fraction; CEST, chemical exchange saturation transfer; APTw, amide proton transfer weighted images; AREX, apparent exchange-dependent relaxation; OE-MRI, oxygen-enhanced MRI; perfused Oxy-R, oxygen-refractory signals in perfused tissue.

**Table 2 cancers-16-02710-t002:** Ongoing trials of magnetic resonance imaging-guided radiation therapy in treatment of lung cancer.

NCT Number (Registration Year)	Study Type	Tumor Type	RT Regimen	Combined Therapy	Trial Design (Arms)	Primary Outcome	Notes
NCT03048760 (2017)	Prospective	Stage I-III NSCLC or SCLC	N/A	Nil	MRI scan	Measure differences between target and OAR volumes contoured on PET, CT, and MRI images	To evaluate the feasibility of MRI for the delineation of OAR and target volumes in lung cancer patients
NCT05237453 (2022)	Prospective	Locally advanced NSCLC	MR-guided ART	Nil	Experimental: MR-guided ART	Clinical feasibility	To demonstrate the feasibility of MR-guided online ART for locally advanced NSCLC
NCT03916419 (2019)	Phase 2	Inoperable stage IIB, IIIA, and select IIIB and IIIC NSCLC	60 Gy/15 Fr	Paclitaxel + Carboplatin + Durvalumab	Chemoradiation + Durvalumab	Safety lead-in only: number of participants with dose-limiting toxicities Phase II only: local control rate; regional control rate	To test the feasibility and outcomes of MR-guided hypofractionated ART with concurrent chemotherapy and consolidation Durvalumab for inoperable stage IIB, IIIA, and select IIIB and IIIC NSCLC
NCT04925583 (2021)	Phase 1	Ultracentral-located lung tumor	SBRT	Nil	Level 0: 10 Fr × 5.0 Gy Level 1: 10 Fr × 5.5 Gy Level 2: 10 Fr × 6.0 Gy Level 3: 10 Fr × 6.5 Gy	Dose-limiting toxicity	To identify the maximum tolerated dose of MR-guided SBRT of ultracentral lung tumors
NCT05354596 (2022)	Phase II	Ultracentrally located lung tumors	SBRT	Nil	MR-Linacs with daily MR-guided plan adaptation	Toxicity: cumulative CTCAE grade ≥ 4 SABR-related toxicity (6, 12, 24, 60 months)	To evaluate the feasibility and safety of daily adaptive MR-Linac-based SBRT in ultracentrally located lung tumors (primary, oligo-metastatic, or oligo-progressive)
NCT05903430 (2023)	Prospective cohort	Centrally located lung cancer	SABR	Nil	Not mentioned	>85% success in delivery and completion of SABR to patients recruited on protocol	To determine if the investigators are able to deliver highly focused, intense radiation to tumors in the abdominal region or chest cavity whilst limiting the dose to OAR using a high-field-strength MR-Linac
NCT04789486 (2021)	Phase 1 Phase 2	Centrally located lung tumors	SMART	AGuIX	Phase 1: AGUIX + SMART Phase 2: AGUIX + SMART; SMART only	Phase 1: MTD Phase 2: compare local control at 12 months of MTD	To help determine the safety and efficacy of gadolinium-based nanoparticle, AGuIX, used in conjunction with SMART in the treatment of pancreatic cancer and lung tumors
NCT04075305 (2019)	Prospective cohort	Cancer patients receiving treatment and/or imaging on an MR-Linac machine	Not mentioned	Nil	Not mentioned	PFS; survival; DFS (3, 6, 24 months) Patient-reported health-related quality of life and tumor-specific quality of life; acute toxicity in CTCAE (3, 6, 12, 24 months) Clinical tumor response; pathological tumor response (2 years)	To facilitate the evidence-based introduction of MR-Linac into clinical practice
NCT04946019 (2021)	Phase 2	Brain metastases from NSCLC	30 Gy/5 Fr	Nil	Experimental: MR-Linac-guided adaptive FSRT	1-year intracranial PFS	To determine the efficacy and safety of MR-Linac-guided adaptive FSRT in patients with brain metastases in NSCLC

NSCLC, non-small-cell lung cancer; SCLC, small cell lung cancer; Gy, gray; Fr, fraction; MRI, magnetic resonance imaging; OAR, organ at risk; PET, positron emission tomography; CT, computed tomography; ART, adaptive radiotherapy; SBRT, stereotactic body radiation therapy; SABR, stereotactic ablative radiotherapy; CTCAE, common toxicity criteria for adverse events; MTD, maximum tolerated dose; AGuIX, Activation and Guidance of Irradiation X; SMART, stereotactic MR-guided adaptive radiation therapy; PFS, progression-free survival; DFS, disease-free survival; FSRT, fractionated stereotactic radiotherapy.

## Data Availability

Data are contained within the article.

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
