# Peer review of "Harnessing the Power of Radiotherapy for Lung Cancer: A Narrative Review of the Evolving Role of Magnetic Resonance Imaging Guidance"

_cancers, 2024, doi:10.3390/cancers16152710_

Round 1
Reviewer 1 Report
Comments and Suggestions for Authors
1. Original Submission
Recommendation to the author and editor:
Major revision
Title: Manuscript ID: cancers-cancers-3102120 entitled "Harnessing the Power of Radiotherapy for Lung Cancer: A Narrative Review of The Evolving Role of Magnetic Resonance Imaging Guidance.
Article Type: Review article
2. Comments to the Corresponding Author:
COPE Ethical guidelines followed during the review process,
The manuscript addresses the magnetic resonance imaging (MRI) implications as it has traditionally played a limited role in lung cancer management compared to computed tomography (CT). Authors described that there is considerable potential for improving the CT-based workflow, especially in visualizing structures like the brachial plexus and chest wall invasion, which are difficult to detect with CT alone. Moreover, in treating high-risk tumors such as ultra-central lung cancer, the toxicity associated with current treatments often outweighs their benefits. Authors described that the introduction of MR-Linac, an MRI-guided radiotherapy (RT) system that combines MRI with a linear accelerator, could address these challenges. MR-Linac offers superior soft tissue visualization, daily adaptive capabilities, real-time target tracking, and early assessment of treatment response compared to CT-based technologies. Clinically, it could be particularly advantageous for treating central/ultra-central lung cancer, early-stage lung cancer, and locally advanced lung cancer. The rising demand for stereotactic body radiotherapy (SBRT) for lung cancer has led to the adoption of MR-Linac in some cancer centers. Authors collected literature to support an overview of the latest developments in imaging-guided radiotherapy (IGRT) with MR-Linac for lung cancer management, highlights advancements in artificial intelligence, and explores new research directions.
Comments:
Overview and general recommendation:
Check for plagiarism for similarity index. Yet, proofreading can enhance the quality of the manuscript. Several sentences need rewriting to make the readers comfortable when reading this. Avoid spelling errors.
1. There is a published report pertinent to this concept on lung cancer [ref: https://www.ncbi.nlm.nih.gov/pmc/articles/PMC6913359/ ], however, MRI was added as a novel point. Can you justify how authors think the addition of MRI guidance without AI would be much beneficial ?
2. Authors should expand the discussion part with additional content. Authors revise the following affiliation: “Ph.D. Program in Environmental and Occupational Medicine, Kaohsiung Medical University and National Health Research Institutes, Kaohsiung, Taiwan”.
3. Cite the following two articles at the appropriate corresponding places inside the manuscript, PMID: 37916622 ; DOI: 10.2174/0113816128226519231017050459 ; doi: 10.3390/metabo12100943
4. As per the previous reports, current AI technologies was able to generate sCT from MR images of the thorax with good dose accuracy and image quality, without inter-scanner variability and radiation exposure. So, I suggest to expand the review by collecting information related to ML (machine learning) models for the Evolving Role of Magnetic Resonance Imaging Guidance & MRgRT Clinical Application with AI technologies ?? Can you add AI sophisticated information into this manuscript ? [ref: https://www.nature.com/articles/s43856-022-00199-0 https://www.nature.com/articles/s41416-019-0412-y , https://www.mdpi.com/2075-4418/13/17/2760 ]
5. I am satisfied with other contents. Authors can enhance the script additional references and Figures to support their review hypothesis.
5. Conclusion should be explained vividly
**Thank you**
Comments on the Quality of English Language
Substantial English language editing is required.
Author Response
COPE Ethical guidelines followed during the review process,
The manuscript addresses the magnetic resonance imaging (MRI) implications as it has traditionally played a limited role in lung cancer management compared to computed tomography (CT). Authors described that there is considerable potential for improving the CT-based workflow, especially in visualizing structures like the brachial plexus and chest wall invasion, which are difficult to detect with CT alone. Moreover, in treating high-risk tumors such as ultra-central lung cancer, the toxicity associated with current treatments often outweighs their benefits. Authors described that the introduction of MR-Linac, an MRI-guided radiotherapy (RT) system that combines MRI with a linear accelerator, could address these challenges. MR-Linac offers superior soft tissue visualization, daily adaptive capabilities, real-time target tracking, and early assessment of treatment response compared to CT-based technologies. Clinically, it could be particularly advantageous for treating central/ultra-central lung cancer, early-stage lung cancer, and locally advanced lung cancer. The rising demand for stereotactic body radiotherapy (SBRT) for lung cancer has led to the adoption of MR-Linac in some cancer centers. Authors collected literature to support an overview of the latest developments in imaging-guided radiotherapy (IGRT) with MR-Linac for lung cancer management, highlights advancements in artificial intelligence, and explores new research directions.
Overview and general recommendation:
Check for plagiarism for similarity index. Yet, proofreading can enhance the quality of the manuscript. Several sentences need rewriting to make the readers comfortable when reading this. Avoid spelling errors.
Responses:
Prior to submission, I have used Turnitin to check for plagiarism and received a similarity index of 4%. (PDF report attached)
Per your request, I have also utilized MDPI language service for proofreading and editing to enhance the quality of the manuscript. (Editing Certificate attached)
- There is a published report pertinent to this concept on lung cancer [ref: https://www.ncbi.nlm.nih.gov/pmc/articles/PMC6913359/ ], however, MRI was added as a novel point. Can you justify how authors think the addition of MRI guidance without AI would be much beneficial ?
Response:
In this article, future developments including both the MR-guided linear accelerator (MRL) and proton beam therapy. The authors believe both represent exciting developments in RT technology which may further advance targeting and individualization of treatment.
The MR-Linac gives better target visualization compared to CBCT and provides the ideal platform to facilitate the investigation of daily online plan adaptation. The MRL combines a linear accelerator with on-board diagnostic quality MRI. The superior soft tissue discrimination of MR gives better target visualization compared to CBCT. The improved target visualization may facilitate the investigation of daily online plan adaptation and isotoxic dose intensification without the concerns of excessive ionizing radiation that exist with CBCT.22,25,59 The MRL will also enable real-time gating and permit assessment of delivered dose compared to planned dose.78 Ultimately, the anatomical and dosimetric information obtained in real time may facilitate intrafractional plan adaptation, to ensure the intended daily dose is delivered. This should permit smaller PTV margins and further reduce the dose to healthy tissues. Therefore, MR-guided RT potentially represents truly individualized RT with a move away from an image-guided to a "dose-guided" technique.22 The MRL also offers the potential to incorporate biological information via functional imaging. This may enable the exciting possibility of individualized adaptation and intensification based on tumour metabolic activity and hypoxia.79
As mentioned in our Section 3.1, Unfortunately, CBCT-generated sCT has the same limitation as planning CT, i.e., poor soft tissue visualizations, and MRgRT could be advantageous in this regard.
I believe that authors think the addition of MRI guidance would be much beneficial even without AI.
I have cited this article in Section 3.3.
The real-time anatomical and dosimetric information can enable intrafractional plan adaptation, ensuring the intended daily dose is delivered. This approach may allow for smaller PTV margins and further reduce the dose to healthy tissues [39].
- Authors should expand the discussion part with additional content. Authors revise the following affiliation: “Ph.D. Program in Environmental and Occupational Medicine, Kaohsiung Medical University and National Health Research Institutes, Kaohsiung, Taiwan”.
Responses: Your opinions on our manuscript are truly appreciated. I have expanded the discussion with additional content as highlighted in yellow.
The affiliation is correct. The Doctoral Degree Program in Environmental and Occupational Medicine was founded in Division of Environmental Health & Occupational Medicine, National Health Research Institutes, and Kaohsiung Medical University in 2010. The National Health Research Institutes (NHRI) is a non-profit foundation established in 1995 in Taiwan. - Cite the following two articles at the appropriate corresponding places inside the manuscript, PMID: 37916622 ; DOI: 10.2174/0113816128226519231017050459 ; doi: 10.3390/metabo12100943
Responses: Thank you very much.
I have cited these two nice articles in the following.
Clinically, there are several methods for MLC calibration or quality assurance [105]
Although the development of these techniques has mitigated side effects to some extent, there are limitations in managing dose distribution [8].
- As per the previous reports, current AI technologies was able to generate sCT from MR images of the thorax with good dose accuracy and image quality, without inter-scanner variability and radiation exposure. So, I suggest to expand the review by collecting information related to ML (machine learning) models for the Evolving Role of Magnetic Resonance Imaging Guidance & MRgRT Clinical Application with AI technologies ?? Can you add AI sophisticated information into this manuscript ?
[ref: https://www.nature.com/articles/s43856-022-00199-0 https://www.nature.com/articles/s41416-019-0412-y , https://www.mdpi.com/2075-4418/13/17/2760 ]
Responses: Thank you. I have added a new section and cited the above papers in the new Section, Section 3.6. AI and machine learning.
3.6. AI and machine learning
AI, especially its subset, machine learning (ML), is revolutionizing radiology by enhancing image analysis and reducing diagnostic errors [74]. AI is transforming radiology by optimizing workflows and improving non-interpretative tasks [75]. Integrated with Natural Language Processing (NLP), AI automates the triage of imaging studies, prioritizing urgent cases through electronic health records, expediting patient triage, reporting, and follow-up management [76]. Deep learning (DL), an ML subset, improves report consistency and clarity, enhancing radiology services quality. DL accelerates MRI scanning, harmonizing efficiency and quality, with commensurate progress witnessed in CT and PET image reconstruction.
Understanding the interrelationships between AI, ML, and DL helps to conceptualize each subfield's contribution and progression within the broader AI narrative [74]. AI provides the foundation, ML enhances AI's potential by enabling machines to learn from data, and DL further extends these capabilities with neural networks that decipher complex data patterns [74]. Despite AI’s potential, only 30% of radiologists used it clinically by 2021, with many skeptical [77]. Effective AI integration requires supportive infrastructure and workflow redefinition. It may help reducing errors and radiologist burnout [78]. Integration of AI into the RT planning pathway will allow ART to become a reality in clinical practice [79]. In one study, AI-generated data were favored over conventional MRI planning data by the radiation oncologist in the study [80]. Accordingly, we believe that the development of AI could speed up the MRgRT workflow and overcome a major barrier to MRgRT implementation.
- I am satisfied with other contents. Authors can enhance the script additional references and Figures to support their review hypothesis.
Responses:
We appreciate your time and efforts. I have added additional references and enhanced both of our originally created figures. Additionally, I have uploaded the originally designed graphic abstract to better support this narrative review.
- Conclusion should be explained vividly
Response:
Thank you for your recommendation. I have re-written the conclusions in a more vivid manner.
MR-Linac is a transformative technology that offers significant advantages over the current CT-based systems in treating lung cancer patients. Its superior soft tissue contrast, daily adaptive capability, and real-time image tracking set it apart. Furthermore, MR-Linac enables the integration of biological and anatomical images into adaptive planning process, enhancing its clinical utility in treating early-stage lung cancer and locally advanced non-small cell lung cancer (LA-NSCLC).
Despite its impressive benefits, MR-Linac does have limitations. As the demand for lung cancer SBRT grows, the adoption of MR-Linac in lung cancer treatment will become more prevalent. Clinicians will gain more experience and data, leading to improved outcomes. To ensure that expediency does not compromise treatment quality or patient safety, future research should focus on optimizing the workflow procedure. AI will play a crucial role in this advancement, enhancing the precision and efficiency of lung cancer treatment with MR-Linac.
Comments on the Quality of English Language
Substantial English language editing is required.
Response:
Thank you for your advises on our manuscript.
I have utilized MDPI language service for proofreading and editing to enhance the quality of the manuscript. (Editing Certificate attached)
We thank you for your time reviewing and giving us dear suggestion on our research. We hope you find the revised manuscript to your satisfaction.
Reviewer 2 Report
Comments and Suggestions for Authors
Although the authors describe the potential benefits of MR-guided radiotherapy (MRgRT) in lung cancer, there remains significant debate concerning the long treatment times and the current lack of respiratory movement modulation, such as that provided by 4D-CT, in MRgRT. These limitations can affect patient comfort, compliance, and overall treatment efficiency. The manuscript should include a more detailed discussion on these potential limitations and address how future advancements could mitigate these issues. Specifically, exploring the integration of advanced respiratory gating technologies or hybrid systems that combine the strengths of MR imaging with 4D-CT capabilities would provide a clearer path forward. Additionally, the authors should discuss ongoing research and development aimed at reducing treatment times, such as automation in treatment planning and AI-driven adaptive radiotherapy processes. Addressing these points would offer a more balanced view and highlight future directions for overcoming the current challenges associated with MRgRT in lung cancer treatment.
Author Response
Although the authors describe the potential benefits of MR-guided radiotherapy (MRgRT) in lung cancer, there remains significant debate concerning the long treatment times and the current lack of respiratory movement modulation, such as that provided by 4D-CT, in MRgRT. These limitations can affect patient comfort, compliance, and overall treatment efficiency. The manuscript should include a more detailed discussion on these potential limitations and address how future advancements could mitigate these issues. Specifically, exploring the integration of advanced respiratory gating technologies or hybrid systems that combine the strengths of MR imaging with 4D-CT capabilities would provide a clearer path forward. Additionally, the authors should discuss ongoing research and development aimed at reducing treatment times, such as automation in treatment planning and AI-driven adaptive radiotherapy processes. Addressing these points would offer a more balanced view and highlight future directions for overcoming the current challenges associated with MRgRT in lung cancer treatment.
Responses:
We agree with you that several limitations to MR-Linac still exist.
1. About long treatment time, I have added some ongoing researches and development aimed at reducing treatment times highlighted in yellow in Section 5. “Challenges for MRgRT in Lung Cancer Treatment”.
Despite its numerous advantages over current CT-based technologies, many limitations related to MRgRT exist. One such challenge is the potential effect of the Lorentz force. During photon irradiation, electrons deposit energy along their pathway, but when these electrons encounter a magnetic field, their path may be deflected by the Lorentz force, which then alters the dose distribution. This is especially significant in areas of heterogenous tissue density such as skin–air or lung–tissue interface regions [10,104]. The Lorentz force at these sites can cause electrons that have already exited the tissue to return to its surface in a phenomenon called the electron return effect (ERE), and, in lung RT, ERE can potentially lead to lung tissue toxicities [10,105]. Fortunately, research so far has not found the dose alteration to be clinically significant [40,105]. Moreover, lung RT usually employs low-field imaging for better imaging quality; thus, the ERE is much lower than in high-field imaging [106].
Clinically, there are several methods for MLC calibration or quality assurance [107]. Some challenges related to the MRgRT workflow include the development of a reliable and consistent quality assurance and quality control procedure that is specific to MRgRT [108]. In addition, there is a relative contraindication to treat patients with metallic implants or cardiac devices. Yang et al. recently reported safely treating four patients with a cardiac pacemaker using a 1.5T MR-Linac [109]; similarly, Keesman et al. also published their MRgRT workflow for three prostate cancer patients with a metallic hip prosthesis [110], although both protocols await confirmation through larger studies. Substantial capital investment and training for team members also impede MR-Linac’s widespread use, and, finally, the low throughput and the time-consuming adaptation process are also major drawbacks of this novel technology. Most importantly, for individuals that have experienced episodes of claustrophobia during an MRI scan, MRgRT may not be suitable.
Moreover, significant debate remains regarding the prolonged treatment times. Such limitations can impact patient comfort, compliance, and overall treatment efficiency. In our experience, from 2020 to 2023, for non-single-fraction BH SMART, the median in-room procedural time was 37 minutes, with a median treatment time of 15 minutes. As discussed in Section 4.2, visual biofeedback respiratory gating was studied by Chuong et al. [95] and reported with good efficiency, and it was examined in another Korean study [96] in which the total treatment time was reduced by 37.6% compared with free-breathing.
Some researchers have taken things further by shortening the entire workflow to one day. A one-step-shop (OSS) service is a procedure well known in the palliative setting that condenses consultation, simulation, and RT delivery all in one day [111] with the goal of shortening overall treatment time, thus improving patient convenience and ameliorating patient discomfort. This is especially important for countries such as the United States and Canada, where patients often must travel great distances to receive RT; therefore, shortening treatment to one day means that time and costs for traveling and lodging could be significantly reduced, further improving the patient adherence rate.
In the past, Brunenberg et al. demonstrated the feasibility of OSS-palliative RT with an MR-based diagnostic protocol and MR simulation [111], and, recently, Palacios et al. applied this procedure to single-fraction SMART for small tumors of early-stage lung cancer [112]. The reported median procedural time was 6.6 hours, and, with procedural optimization, the treatment could be completed in half a day. Good patient satisfaction was reported, with highly accurate gated treatment delivery, as demonstrated by the above 78 -100% GTV coverage and 94.4–100% corresponding PTV values. Taken together, these studies demonstrate that MRgRT is capable of performing single-fraction SBRT with high precision, but plenty of room remains for discussing the exact workflow and implementation.
However, as we previously discussed, research efforts are being made to ameliorate these problems. The PUMA trial (NCT05237453) is a prospective, multicenter phase I trial in Germany that will test the clinical feasibility of the two MR-Linac systems in CCRT treatment for LA-NSCLC patients [113]. It will use conventionally fractionated RT with dose escalation and weekly online adaptation. Their primary end points include the successful completion of online-adapted fractions and on-table time. Additionally, AI development and procedural optimization as team members become more familiar with the new workflow can significantly increase throughput over time. For example, the manual delineation of OARs represents one of the most time-consuming processes of MRgRT, and AI technologies that can perform both left and right lung auto-segmentations with high accuracy have already been developed [80].
Regarding 4D-CT,
in Section 3.3 Real-Time Target Tracking, we elaborated about 4D-CT.
For thoracic imaging and RT, a major source of intrafractional motion is respiratory motion. Currently, respiratory motions are accommodated through the expansion of the clinical target volume (CTV)–planning target volume (PTV) margin. It is important to note that, although 4D-CT could account for structural changes with time, it does not account for real-time daily motion management. This is because 4D-CT is conducted at simulation, where the machine studies tumor motion through a patient’s respiratory cycle and calculates an internal target volume (ITV) based on the union of all tumor positions during the imaging process; however, 4D-CT does not account for the changes that occur during RT delivery. In fact, a study using real-time MRI found that ITV size often varies more than previously considered, suggesting significantly highernter- and intra-fractional variations in breathing amplitude than accounted for during the current ITV-based planning process [38]. The real-time anatomical and dosimetric information can enable intrafractional plan adaptation, ensuring the intended daily dose is delivered. This approach may allow for smaller PTV margins and further reduce the dose to healthy tissues [39].
- Regarding AI, I have written a new Section
3.6. AI and machine learning
AI, especially its subset, machine learning (ML), is revolutionizing radiology by enhancing image analysis and reducing diagnostic errors [74]. AI is transforming radiology by optimizing workflows and improving non-interpretative tasks [75]. Integrated with Natural Language Processing (NLP), AI automates the triage of imaging studies, prioritizing urgent cases through electronic health records, expediting patient triage, reporting, and follow-up management [76]. Deep learning (DL), an ML subset, improves report consistency and clarity, enhancing radiology services quality. DL accelerates MRI scanning, harmonizing efficiency and quality, with commensurate progress witnessed in CT and PET image reconstruction.
Understanding the interrelationships between AI, ML, and DL helps to conceptualize each subfield's contribution and progression within the broader AI narrative [74]. AI provides the foundation, ML enhances AI's potential by enabling machines to learn from data, and DL further extends these capabilities with neural networks that decipher complex data patterns [74]. Despite AI’s potential, only 30% of radiologists used it clinically by 2021, with many skeptical [77]. Effective AI integration requires supportive infrastructure and workflow redefinition. It may help reducing errors and radiologist burnout [78]. Integration of AI, such as automation into the RT planning pathway will allow ART to become a reality in clinical practice [79]. In one study, AI-generated data were favored over conventional MRI planning data by the radiation oncologist in the study [80]. Accordingly, we believe that the development of AI-driven ART could speed up the MRgRT workflow and overcome a major barrier to MRgRT implementation.
I appreciate that you have given me valuable suggestions. I hope you find the revised manuscript to your satisfaction.
Thank you very much.
Round 2
Reviewer 1 Report
Comments and Suggestions for Authors
Authors addressed my comments adequately. You can accept. But I suggest authors to proofread at least 3 to 4 times before publication to avoid conceptual errors.